# Optimized Hybrid Deep Learning Framework for Early Detection of Alzheimer’s Disease Using Adaptive Weight Selection

**DOI:** 10.3390/diagnostics14242779

**Published:** 2024-12-11

**Authors:** Karim Gasmi, Abdulrahman Alyami, Omer Hamid, Mohamed O. Altaieb, Osama Rezk Shahin, Lassaad Ben Ammar, Hassen Chouaib, Abdulaziz Shehab

**Affiliations:** 1Department of Computer Science, College of Computer and Information Sciences, Jouf University, Sakaka 72388, Saudi Arabia; kgasmi@ju.edu.sa (K.G.); moeltaib@ju.edu.sa (M.O.A.); orshahin@ju.edu.sa (O.R.S.); 2Department of Information Systems, College of Computer and Information Sciences, Jouf University, Sakaka 72388, Saudi Arabia; aishehab@ju.edu.sa; 3Cybersecurity Department, College of Engineering and Information Technology, Buraydah Private Colleges, Buraydah 51418, Saudi Arabia; omer.abdullah@bpc.edu.sa; 4College of Computer Engineering and Sciences, Prince Sattam bin Abdulaziz University, Al-Kharj 11942, Saudi Arabia; l.benammar@psau.edu.sa; 5Department of Physics, College of Science, Jouf University, P.O. Box 2014, Sakaka 72341, Saudi Arabia; hchouaib@ju.edu.sa

**Keywords:** Alzheimer detection, ensemble learning, optimal algorithm

## Abstract

Background: Alzheimer’s disease (AD) is a progressive neurological disorder that significantly affects middle-aged and elderly adults, leading to cognitive deterioration and hindering daily activities. Notwithstanding progress, conventional diagnostic techniques continue to be susceptible to inaccuracies and inefficiencies. Timely and precise diagnosis is essential for early intervention. Methods: We present an enhanced hybrid deep learning framework that amalgamates the EfficientNetV2B3 with Inception-ResNetV2 models. The models were integrated using an adaptive weight selection process informed by the Cuckoo Search optimization algorithm. The procedure commences with the pre-processing of neuroimaging data to guarantee quality and uniformity. Features are subsequently retrieved from the neuroimaging data by utilizing the EfficientNetV2B3 and Inception-ResNetV2 models. The Cuckoo Search algorithm allocates weights to various models dynamically, contingent upon their efficacy in particular diagnostic tasks. The framework achieves balanced usage of the distinct characteristics of both models through the iterative optimization of the weight configuration. This method improves classification accuracy, especially for early-stage Alzheimer’s disease. A thorough assessment was conducted on extensive neuroimaging datasets to verify the framework’s efficacy. Results: The framework attained a Scott’s Pi agreement score of 0.9907, indicating exceptional diagnostic accuracy and dependability, especially in identifying the early stages of Alzheimer’s disease. The results show its superiority over current state-of-the-art techniques.Conclusions: The results indicate the substantial potential of the proposed framework as a reliable and scalable instrument for the identification of Alzheimer’s disease. This method effectively mitigates the shortcomings of conventional diagnostic techniques and current deep learning algorithms by utilizing the complementing capabilities of EfficientNetV2B3 and Inception-ResNetV2 by using an optimized weight selection mechanism. The adaptive characteristics of the Cuckoo Search optimization facilitate its application across many diagnostic circumstances, hence extending its utility to a wider array of neuroimaging datasets. The capacity to accurately identify early-stage Alzheimer’s disease is essential for facilitating prompt therapies, which are crucial for decelerating disease development and enhancing patient outcomes.

## 1. Introduction

Alzheimer’s disease (AD) is a devastating neurological disorder that gradually impairs memory, thinking, and the ability to carry out simple tasks. Often beginning with mild forgetfulness, AD progressively leads to severe cognitive decline, affecting millions of people worldwide. As of 2020, more than 5.8 million Americans were living with Alzheimer’s disease, mostly older adults. Globally, the number is expected to skyrocket in the coming decades, making AD not just a personal tragedy for those affected and their families but also a massive societal challenge. Caring for people with AD strains healthcare systems, caregivers, and the economy. Early diagnosis is critical for managing symptoms and improving quality of life, but it remains one of the most complex aspects of the disease.

One of the biggest hurdles in diagnosing Alzheimer’s disease is that its symptoms often overlap with normal aging or other forms of dementia, especially in its early stages. While cognitive tests and memory assessments are frequently used, they’re often not definitive. More accurate methods, such as brain imaging techniques like MRI, can detect changes in brain structure associated with AD. However, these procedures are expensive and not always available in all healthcare settings, particularly in underserved regions. As a result, there’s a growing demand for more accessible, efficient diagnostic tools that can provide earlier and more reliable detection of AD.

Artificial intelligence (AI) [1] and machine learning are emerging as powerful tools in the fight against Alzheimer’s disease [2,3]. By training models to recognize patterns in brain images or other biomarkers [4], these technologies hold the promise of automating the diagnosis process, potentially making it faster and more accurate. In particular, deep learning models [5,6,7,8], like convolutional neural networks (CNNs), have shown great promise in analyzing medical images to detect disease patterns. However, these models have limitations: they often require vast amounts of data and computational power, and they can sometimes struggle to generalize across different patient populations or datasets. This is where combining multiple models, an approach known as ensemble learning, can come into play.

Ensemble learning involves using multiple AI models to make predictions and then combining their outputs to create a more accurate final decision. By blending the strengths of different models, this method can reduce errors and improve performance, especially in complex tasks like diagnosing Alzheimer’s disease. Though ensemble learning has been explored in other areas of healthcare, such as cancer detection, it is less commonly applied to AD diagnosis.

This study aims to develop an optimal ensemble learning system for diagnosing Alzheimer’s disease using neuroimaging data from the well-known Alzheimer’s Disease Neuroimaging Initiative (ADNI) database. The focus is on integrating multiple deep learning models, improving the way features from these models are selected and combined, and fine-tuning how the different models’ predictions are weighted to ensure the most accurate results.

The paper is organized as follows: Section 2 reviews related work, Section 3 details the materials and methods employed in this research, Section 4 analyzes the experimental results, and Section 5 concludes the study.

## 2. Related Work for Alzheimer Disease Detection

In recent years, the application of machine learning and deep learning techniques to Alzheimer’s disease detection has gained significant attention, driven by the need for early diagnosis and intervention. Various machine learning methods have been employed to analyze neuroimaging and clinical data, offering promising results in predicting disease progression. With its ability to automatically extract complex features, Deep learning has further advanced the accuracy of Alzheimer’s disease detection models. More recently, ensemble learning approaches, which combine multiple models to enhance prediction performance, have shown great potential in improving diagnostic accuracy by leveraging the strengths of different algorithms. The following sections will explore key studies in each of these areas.

### 2.1. Machine Learning-Based Alzheimer’s Disease Detection

Machine learning has become a crucial tool in the effort to detect Alzheimer’s disease earlier and more accurately. Instead of relying solely on human analysis, machine learning algorithms are designed to process large volumes of data such as MRI scans, genetic profiles, and clinical records to find patterns that may signal the onset of Alzheimer’s disease.

Despite the advances machine learning brings, it faces certain limitations. One of the primary challenges is the need for large, labeled datasets to train these models effectively, which can be difficult to obtain in medical research. Furthermore, while machine learning models can provide accurate predictions, understanding how they arrive at these conclusions can be challenging. This “black box” problem makes it difficult for healthcare professionals to fully trust the results without clear explanations, which is essential in medical decision-making. Nonetheless, the continued development of machine learning approaches holds great promise for improving Alzheimer’s disease diagnosis, offering new tools to help doctors detect the disease earlier and more reliably.

There is a growing body of literature demonstrating the successful application of machine learning to Alzheimer’s disease diagnosis. For instance, in [9], the authors proposed a model that utilizes artificial intelligence to predict the progression from Mild Cognitive Impairment (MCI) to Alzheimer’s disease (AD) within 1 year. Their model, based on convolutional neural networks (CNNs) for feature extraction and support vector machines (SVMs) for classification, achieved a classification accuracy of up to 92.3%. Similarly, a hybrid approach combining 3D CNNs with SVMs for early detection of Alzheimer’s disease was introduced in [10], with the model achieving a classification accuracy of 91%, highlighting the efficacy of SVMs in medical image classification.

Another study integrated Kernel Principal Component Analysis (KPCA) with SVM and utilized Discrete Wavelet Transform (DWT) for feature extraction, resulting in high accuracy in classifying MRI images as normal or abnormal. The use of k-fold cross-validation further demonstrated elevated performance in terms of accuracy, specificity, and sensitivity. An approach employing genetic algorithms and SVMs for Alzheimer’s disease diagnosis was presented in [11], where MRI data showing early signs of cognitive disorders were classified into Alzheimer’s disease stages. This method reported a feature detection accuracy of 96.80%, a recall of 89.13%, and a precision of 93.01%.

A novel approach for feature selection using particle swarm optimization combined with a weighted SVM was developed in [12], achieving a classification accuracy of 93%. This technique demonstrated its ability to handle large datasets and provided promising results in Alzheimer’s disease diagnosis. In [13], the importance of using multiple imaging modalities, including MRI, CT, EEG, and PET, for machine learning-based Alzheimer’s disease diagnosis was discussed, highlighting the potential of these techniques in identifying early-stage Alzheimer’s disease with high precision.

Another study [14] utilized SVM in conjunction with feature extraction from MRI brain slices using Haralick features and the Grey Level Co-occurrence Matrix (GLCM), achieving an accuracy of 84% in distinguishing Alzheimer’s disease patients from healthy controls. This method underscores the potential of texture-based feature extraction in improving early detection.

In [15], a CNN-based model classified brain MRI data from the ADNI dataset into four categories: MCI, AD, cognitively normal (CN), and healthy individuals, achieving an accuracy of 95.82%. The study encountered limits due to constraints in computational resources and operational difficulties, especially when utilizing cloud-based platforms such as Google Colab. The emphasis on Alzheimer’s disease-specific data may restrict the model’s applicability to wider medical categorization tasks.

Additionally, random forest algorithms have been applied to feature identification and extraction in Alzheimer’s disease diagnosis [16]. This method, which achieved a success rate of 93%, demonstrates the potential of decision tree-based classifiers in medical image analysis. The study in [17] explored morphometric methods for Alzheimer’s disease diagnosis, employing SVM for feature extraction alongside Principal Component Analysis (PCA) and Independent Component Analysis (ICA) for variable selection. This approach yielded an accuracy of 95%.

### 2.2. Deep Learning-Based Alzheimer’s Disease Detection

The use of deep learning in the detection of Alzheimer’s disease has gained significant attention in recent years due to its ability to automatically analyze and interpret complex medical data, particularly neuroimaging.

Despite the promising results, there are still challenges to overcome. Deep learning models require large, well-labeled datasets to achieve high accuracy, and the integration of different types of data, such as combining MRI scans with clinical records, can be difficult. Additionally, ensuring that these models are interpretable by clinicians and generalizable across different populations remains an ongoing area of research. Nonetheless, the potential of deep learning to revolutionize Alzheimer’s disease diagnosis is clear, offering hope for earlier and more accurate detection in the future.

The authors in [18] utilized the ADNI dataset to implement ResNet and GoogleLeNet models, devising a four-way classifier to differentiate between Alzheimer’s disease (AD), Mild Cognitive Impairment (MCI), Late MCI (LMCI), and Cognitively Normal (CN) participants. Their findings indicated that GoogleLeNet outperformed ResNet, achieving a prediction accuracy of 98.8%. However, the system’s robustness was limited due to the absence of clinical data and insufficient multi-modal data management, which constrained the model’s overall performance.

A 2018 study [19] introduced cascaded CNNs for analyzing MRI and PET brain images from the ADNI dataset, comprising 100 NC, 93 AD, and 204 MCI individuals. This method did not necessitate specialist skills or picture segmentation. The multi-modal analysis achieved an accuracy of 95.68%, although the requirement for an extensive training dataset presented obstacles for practical use. The absence of clinical data integration and multi-modal management further constrained the model’s resilience.

The authors of [20] enhanced the VGG architecture for multi-class classification of Alzheimer’s disease by employing fine-tuned pre-trained CNNs for feature extraction. This model attained exceptional categorization accuracy.Despite its success, the model faced constraints when trained with 64 images per participant, leading to the introduction of superfluous or noisy data. To mitigate this, the data were reduced to 32 images per person to enhance accuracy.

In [21], the authors proposed a Siamese CNN (SCNN) based on the VGG-16 architecture, utilizing two parallel VGG-16 layers for dementia stage classification. By applying data augmentation techniques, they enhanced the diversity of the OASIS dataset, achieving good test accuracy. Nevertheless, the study was hindered by the scarcity of annotated data and the complexity of model parameters, which limited its overall accuracy.

Orouskhani et al. [22] proposed a unique deep triplet network for detecting Alzheimer’s disease using brain MRI data. Due to the limited dataset, a conditional loss function was incorporated to improve the model’s accuracy, demonstrating the potential of metric learning in this domain.

In [23], the authors explored the use of ResNets to predict the progression from MCI to AD. The model aimed to identify individuals with MCI who would progress to AD within 3 years. Despite reaching a test classification accuracy of 83.01%, the study was limited by insufficient training data and the high computational costs associated with training deep CNNs.

The research in [24] introduced a revised iteration of AlexNet, termed Alzheimer’s Disease Detection Empowered with Transfer Learning (ADDTLA), for the identification of Alzheimer’s disease across four stages: moderate dementia (MoD), mild dementia (MD), very mild dementia (VMD), and non-dementia (ND). The model attained 91.7% accuracy in early-stage diagnosis; nevertheless, it did not undergo a comprehensive assessment of precision and did not optimize all convolutional layers, which may restrict its performance. The study also did not investigate other pre-trained networks or CNN designs that could have enhanced the outcomes.

In [25], the authors employed CNNs on MRI image datasets to classify participants into CN, MCI, Early MCI (EMCI), and LMCI categories. They utilized transfer learning with pre-trained models like MobileNet, achieving 96% accuracy in multi-class Alzheimer’s disease classification. While other models, such as VGG16 and ResNet50, were also tested, the lack of data augmentation techniques like downsampling may have affected the model’s robustness. Furthermore, the study did not explore multi-modal fusion methods, which could have provided a more comprehensive approach to Alzheimer’s disease classification.

### 2.3. Ensemble Learning-Based Alzheimer’s Disease Detection

Ensemble learning is based on the idea that combining multiple models, known as “weak learners”, can lead to a stronger, more accurate prediction. Instead of relying on a single model, ensemble methods combine the strengths of various models to produce better results.

In Alzheimer’s disease detection, ensemble learning is particularly useful because the problem is complex and varies from patient to patient. A single model may not always capture the full range of data patterns or handle the intricacies involved in analyzing brain scans or clinical information.

This hybridization allows for a more accurate and resilient prediction, helping to better tackle the challenges associated with diagnosing Alzheimer’s disease from large and diverse datasets.

Ensemble learning has emerged as a powerful tool for improving the accuracy of Alzheimer’s disease detection by combining the strengths of multiple models. One study utilized an ensemble of Deep Convolutional Neural Networks (DCNNs) on MRI scans from the OASIS dataset, showing that models like Inception-v4 and ResNet performed well due to better gradient flow during training. The proposed model, which combined three different models (M1, M2, and M3), reached an accuracy of 93.18% despite the relatively small dataset, although the model’s complexity required a larger dataset for optimal performance [26].

In a different method, ensemble learning was paired with the MDR constructive induction algorithm in order to find complicated genetic relationships connected to Alzheimer’s disease. This was a significant step toward the development of customized therapy. This framework used techniques like Random Forest (RF), XGBoost, and Classification and Regression Trees (CARTs) to enhance the detection of these interactions [27].

Ruiz et al. [28] employed an ensemble of 3D DenseNet models to classify MRI images into four categories, utilizing dense connections that improved data flow throughout the model. Similarly, Pan et al. [29] proposed a CNN and ensemble learning method (CNN-EL) to classify individuals with MCI or AD based on MRI slices from different planes, leading to a notable improvement in accuracy.

Razzak et al. [30] introduced PartialNet, an ensemble specifically designed for Alzheimer’s disease detection using MRI. PartialNet incorporates identity mappings and varied model depths, allowing for better feature reuse and enhancing the system’s ability to learn efficiently.

Other ensemble-based approaches include the work by Yiru et al. [31], who developed a model combining CNNs and LSTM networks, using knowledge distillation to reduce computational costs while achieving an accuracy of 85.96%. Maysam et al. [22] addressed the issue of limited data by using a conditional deep triplet network built on the VGG-16 architecture, which significantly improved classification performance on Alzheimer’s disease datasets.

Hazarika et al. [32] used DenseNet-121 for Alzheimer’s disease classification and later proposed a hybrid model combining LeNet and AlexNet, achieving an accuracy of 93.58%. Fu’adah et al. [33] employed a CNN based on AlexNet to automate the classification of Alzheimer’s disease stages, with an accuracy of 95%, highlighting the effectiveness of ensemble learning in assisting healthcare professionals with more accurate diagnosis and treatment planning.

## 3. Proposed Model for Alzheimer Detection

In this study, we propose an approach, Figure 1, that combines two deep learning models at a time to leverage the strengths of each architecture. Our approach is particularly designed for Alzheimer’s disease detection using medical imaging data, such as MRI scans. The goal is to improve classification accuracy and model robustness by harnessing the complementary features learned by different models.

To achieve this, we integrated a weight selection method based on an optimization algorithm. In our approach, each model contributes a probability to the final classification, and the optimization algorithm assigns a weight to each model’s output. These weights are dynamically optimized during training to ensure that the most reliable model in each pair contributes more to the final decision. This allows for a more flexible and adaptive classification system, which is crucial for complex tasks like Alzheimer’s disease detection, where subtle differences in medical images can make a significant impact on diagnostic outcomes.

The optimization algorithm we use ensures that the selected weights for each pair of models are optimal in terms of minimizing classification error. The models are combined in a way that the weighted probabilities from both models produce a final prediction, enhancing the overall performance. For example, when EfficientNetB0 is paired with ResNet50, the optimization algorithm ensures that each model’s contribution to the final decision is weighted according to its performance on the task, leading to improved diagnostic accuracy.

This method of combining models and dynamically adjusting their contributions offers a novel way to handle classification tasks, particularly in medical applications like Alzheimer’s disease detection. The integration of the weight selection method via optimization provides a more tailored and precise classification system capable of adapting to the complexity and variability of medical data.

### 3.1. Dataset and Data Pre-Processing

The OASIS (Open Access Series of Imaging Studies) (https://www.oasis-brains.org/ (accessed on 15 August 2024)) dataset is a widely used collection of brain imaging data and is primarily used to focus on understanding and diagnosing Alzheimer’s disease. The dataset contains structural brain MRI scans, along with clinical and demographic information, and has been instrumental in advancing research in Alzheimer’s disease detection and neuroimaging.

The dataset, sourced from the OASIS study, consists of 80,000 MRI brain scans taken along the z-axis and includes data from 461 participants. The distribution of scans across the severity levels shows a significant class imbalance, with 67,222 scans in the ‘none’ category, 13,725 in ‘very mild’, 5002 in ‘mild’, and 488 in ‘moderate’. Figure 2 present same examples taken from the dataset.

There is a noticeable class imbalance in the dataset presented in Figure 3, where some categories have far more samples than others. This imbalance can create challenges for deep learning models, as they tend to favor the majority class, making it harder for the model to correctly identify and classify cases in the minority classes. To develop a reliable and fair model, it is essential to address this imbalance effectively.

To address the class imbalance [34], methods like oversampling the minority class, undersampling the majority class, or using more advanced techniques like the Synthetic Minority Over-sampling Technique (SMOTE) can be applied to improve the model’s ability to learn from the under-represented classes. In our case, we experimented with several approaches and ultimately opted for a combination of over- and undersampling.

Furthermore, it is important to use evaluation metrics that reflect the impact of class imbalance. Metrics like precision, recall, and the F1-score provide a more accurate assessment of model performance compared to relying solely on accuracy.

#### Data Preparation and Handling Class Imbalance


*
**Step1: Splitting the Test Set**
*


Before starting any pre-processing and applying oversampling techniques, we first set aside a test set of images from each category. This ensures that the model is evaluated on completely unseen data, giving a more accurate reflection of its performance. Importantly, this test set was not involved in hyperparameter tuning, so it offers an unbiased assessment of the model’s capabilities.

Specifically, 20% of the “moderate dementia” samples and 770 images from each of the other three categories were reserved to form the test set. Although the test set was not proportionally distributed across classes, this approach reflects real-world diagnostic conditions, where certain categories, such as “moderate dementia”, are naturally under-represented. This design ensures that the test set better mirrors realistic clinical scenarios and predictions.

This approach ensured that the training and validation datasets were balanced, facilitating effective model learning, and the test set remained independent and comprised only real images. This separation was critical to provide an unbiased evaluation of the model’s performance.


*
**Step 2: Addressing Class Imbalance**
*


Our dataset was highly imbalanced, with the smallest class representing only 0.007% of the largest class. While this imbalance mirrors the actual distribution of Alzheimer’s disease cases (with more severe stages being rarer), it poses a significant challenge for training a neural network. A model trained on imbalanced data can show high accuracy for the majority class but struggle with under-represented categories, resulting in misleading performance metrics like accuracy. Handling this imbalance was a key priority.

A particular complication in this case is that we are working with medical brain scans. Common data augmentation techniques, such as rotations or distortions, are not advisable for medical data since altering these images can remove important structural information [35], potentially leading to inaccurate diagnoses. Given the ethical implications, we avoided aggressive augmentation methods.

We explored several solutions, including undersampling and oversampling. Ultimately, we found that a combination of undersampling the majority classes and oversampling the minority classes provided the best results, which we used in our final model.

For the training and validation sets, we undersampled the majority classes and oversampled the minority ones to achieve 6000 images per class. This gave us a balanced dataset of 24,000 images, which we split into 19,000 for training and 5000 for validation. This balanced set helped train the model effectively while avoiding the pitfalls of class imbalance.

### 3.2. Deep Learning Model

In our hybrid learning approach, we implemented six distinct models: EfficientNetB0, EfficientNetB3V2, ResNet50, Inception-ResNet50, MobileNet, and ConvNet to enhance Alzheimer’s disease detection. Each model brings unique strengths to feature extraction and image classification, allowing us to capture varying levels of detail in MRI brain scans. By combining these models, we aimed to improve diagnostic accuracy and robustness across multiple disease stages.

#### 3.2.1. EfficientNetB0

EfficientNetB0 [36] is a groundbreaking model that strikes an impressive balance between performance and efficiency. What sets it apart from traditional architectures is its use of compound scaling—a method that scales the depth, width, and resolution of the network in a balanced way. Instead of just making the network deeper or wider, EfficientNetB0 grows in a thoughtful, balanced manner that allows it to achieve great accuracy without consuming too much computational power. This makes it ideal for tasks where speed and efficiency are critical, such as medical imaging on resource-constrained systems. By including EfficientNetB0 in our model combinations, we benefit from its ability to handle tasks that require precision with minimal overhead, which is particularly useful when working with high-resolution scans in Alzheimer’s disease detection.

EfficientNet models are based on compound scaling, which systematically scales the model’s width, depth, and resolution. The scaling formula is
(1)$$d=\alpha^{\phi },\;w=\beta^{\phi },\;r=\gamma^{\phi }$$
where

-*d* is the depth (number of layers);-*w* is the width (number of channels per layer);-*r* is the image resolution;-α,β, and γ are constants that control how depth, width, and resolution are scaled, respectively;-ϕ is the scaling coefficient, which controls overall scaling based on available resources.

This equation ensures a balanced increase in network size while maintaining efficiency.

#### 3.2.2. EfficientNetB3V2

A more advanced version of the EfficientNet family [37], EfficientNetB3V2, refines the strengths of its predecessor, incorporating newer techniques like batch normalization and enhanced scaling to boost performance. It is a more powerful model designed to manage higher computational loads while still maintaining the trademark efficiency of the EfficientNet series. For complex image classification tasks, like those needed in Alzheimer’s detection, EfficientNetB3V2 provides superior accuracy while remaining computationally light enough to be practical in real-world applications. By pairing it with other models in our approach, we harness its ability to work through intricate details in the data, offering a deeper layer of analysis without sacrificing speed.

#### 3.2.3. ResNet50

ResNet50 is one of the most iconic models in deep learning, known for its innovative use of residual connections. These connections help the model avoid common training pitfalls, like the vanishing gradient problem, by allowing layers to skip connections and pass information directly across the network. This makes ResNet50 extremely powerful for extracting deep, hierarchical features from images. In the context of Alzheimer’s disease detection, ResNet50 can capture subtle, deep patterns in MRI or CT scans that may be missed by shallower networks. By combining ResNet50 with other models, we tap into its ability to dig deep into the data and find patterns that are crucial for accurate diagnosis, giving our approach a more thorough feature extraction process [38].

ResNet50 uses residual connections to avoid the vanishing gradient problem by skipping layers. The mathematical expression for a residual block is
(2)$$y=F\left(x,\left\{W_{i}\right\}\right)+x$$
where

-*x* is the input to the residual block;-$F\left(x,\left\{W_{i}\right\}\right)$ is the learned residual mapping (the result of the convolution layers);-*y* is the output of the block.

These residual connections allow the network to train deeper layers effectively.

#### 3.2.4. Inception-ResNet50

The Inception-ResNet50 model is an exciting fusion of two powerful architectures: Inception and ResNet. The Inception component brings the ability to process features at multiple scales simultaneously, while the ResNet component ensures that the model can be trained to significant depth without running into problems like vanishing gradients. This combination makes Inception-ResNet50 particularly adept at tackling complex visual tasks, where both fine details and broader contextual information are important. In our Alzheimer’s disease detection framework, this model helps by capturing a range of features, from the smallest anomalies to more generalized patterns in the brain scans, adding richness and depth to our predictions [39].

#### 3.2.5. MobileNet

MobileNet is all about lightweight efficiency. Designed for use in mobile and embedded systems, it uses a clever technique called depthwise separable convolutions to reduce the number of parameters and computations while still achieving impressive accuracy. This makes it perfect for real-time applications or settings where computational resources are limited. In our combined model approach, MobileNet brings speed and efficiency to the table, offering a quick and effective way to process large datasets, especially when used alongside heavier models like ResNet50. This combination allows us to build a more adaptive and responsive system for Alzheimer’s disease detection, capable of providing fast preliminary results without compromising accuracy [40].

MobileNet uses depthwise separable convolutions, which significantly reduce computation by splitting the convolution operation into two parts. The depthwise separable convolution is given by
(3)$$Y=DW\left(X,K_{d}\right)\times PW\left(K_{p}\right)$$
where

-*X* is the input;-$K_{d}$ and $K_{p}$ are depthwise and pointwise convolution filters, respectively;-DW is the depthwise convolution (operating separately on each channel);-PW is the pointwise convolution (combining outputs from depthwise convolution).

#### 3.2.6. ConvNet (Convolutional Neural Network)

The convolutional neural network (ConvNet) is the classic workhorse of deep learning, particularly well-suited to image data. ConvNets are composed of layers of filters (or “kernels”) that automatically learn to detect patterns in image edges, textures, shapes, and beyond. What makes ConvNets so powerful is their ability to capture spatial hierarchies in the data, which is crucial for understanding images at both the local and global levels. In our approach, ConvNet serves as a foundational model, offering reliable feature extraction and complementing the more sophisticated architectures we pair it with. Its simplicity and proven effectiveness make it an essential part of our toolset for diagnosing Alzheimer’s disease from complex medical images [41].

A standard convolutional layer in ConvNet can be mathematically expressed as
(4)$$y_{i,j,k}=\sum_{m}\sum_{n}x_{i+m,j+n}\cdot w_{m,n,k}+b_{k}$$
where

-*x* is the input image or feature map;-*w* represents the learned weights (filters);-$b_{k}$ is the bias for filter *k*;-*y* is the output feature map.

### 3.3. Combination of Two Models: Average and Weighted Average Techniques

In our proposed approach, we first explore the simple average combination technique, followed by an enhanced method based on a weighted average, where the weights are dynamically selected by an optimization algorithm to improve the overall performance.


*
**Step 1: Average Combination Technique**
*


The initial step in our approach is to combine the outputs of two models using the simple average method. In this technique, both models contribute equally to the final classification result. The probabilities predicted by each model are added together and then divided by two, giving equal importance to both. For instance, when combining Model A and Model B to classify Alzheimer’s disease, the final probability output $P_{\mathrm{final}}$ would be calculated as
(5)$$P_{\mathrm{final}\;}=\frac{P_{A}+P_{B}}{2}$$

This method is easy to implement and can be quite effective, especially when the models perform similarly or have complementary strengths. However, the limitation is that it assumes both models are equally reliable, which may not always be the case, particularly in complex medical tasks like Alzheimer’s detection, where subtle variations in the data can lead to differences in model performance.


*
**Step 2: Weighted Average Combination based on a Selection Technique:**
*


To improve on the simple average method, we introduce a weighted average combination, where each model’s contribution to the final decision is weighted based on its performance. The weights for each model are not static; instead, they are selected dynamically by an optimization algorithm that minimizes the classification error.

In this approach, the output from each model is multiplied by a weight, and the final classification is based on the weighted sum of the predictions. The formula for the weighted average combination is
(6)$$P_{\mathrm{final}\;}=w_{A}\cdot P_{A}+w_{B}\cdot P_{B}$$

Here, $w_{A}$ and $w_{B}$ represent the weights assigned to Model A and Model B, respectively. The optimization algorithm adjusts these weights during the training process to ensure that the final combined model performs optimally.

The weighted average method offers several advantages. It allows the model that performs better to have more influence on the final decision, while the less accurate model contributes less. This approach is particularly useful in tasks like Alzheimer’s disease detection, where different models may excel at identifying different features in medical images. By optimizing the weights, we ensure that the final model adapts to the data’s complexity and delivers more accurate classifications.

In summary, by starting with a simple averaging approach and then enhancing it with a dynamic, optimized weighted average method, we create a more flexible and precise model. This not only improves the overall performance but also makes the system more adaptable to the unique challenges of Alzheimer’s disease detection.

### 3.4. Weight Selection Using Cuckoo Search for Combining Deep Learning Models

In this study, we use the Cuckoo Search (CS) algorithm to select the optimal weights when combining the outputs of two deep learning models for Alzheimer’s disease detection. Cuckoo Search is a powerful optimization technique inspired by the natural behavior of cuckoo birds, which lay their eggs in the nests of other birds. This method has gained popularity due to its ability to efficiently explore the solution space while avoiding local optima, making it well-suited for weight optimization in deep learning models, especially in complex tasks like medical image classification [42,43].

The final prediction from our combined models is represented by Equation (Equation 6).

#### 3.4.1. Cuckoo Search Algorithm Overview

Cuckoo Search uses random walks known as Lévy flights to search through the solution space. These random steps allow the algorithm to cover a wide area, increasing the chances of finding the optimal solution without getting stuck in sub-optimal regions [42]. The basic principles of CS can be summarized as the following:

***Egg Laying:*** Each cuckoo lays one egg (a new potential solution) in a randomly chosen nest (a solution candidate).

***Fitness Evaluation:*** The quality of the egg (solution) is evaluated; in our case, this is based on the classification accuracy achieved using the current weight combination.

***Nest Replacement:*** The best nests (solutions) are retained for the next iteration, while a portion of the less optimal ones are replaced with new solutions, allowing the algorithm to explore new regions of the solution space [44].

#### 3.4.2. Applying Cuckoo Search for Weight Selection

For our task of combining two deep learning models, we followed a structured process to optimize the weights assigned to each model’s prediction. Here is how we apply Cuckoo Search:

***Initialization:*** A population of candidate weight combinations $\left(w_{A},w_{B}\right)$ is initialized. Each combination represents a possible way of combining the outputs of the two models.

***Fitness Evaluation:*** For each candidate combination of weights, we compute the final prediction, $P_{\mathrm{final}}$, and evaluate its accuracy on the validation set. The accuracy serves as the fitness score for each weight combination [45].

***Lévy Flights for Exploration:*** Lévy flights enable the algorithm to take both large and small steps, which helps in searching the solution space efficiently. This ensures that the algorithm does not get stuck in local optima [42,46].

***Selection and Replacement:*** After evaluating the fitness of all solutions, the best-performing weight combinations are retained, and a portion of lower-performing solutions is replaced. This maintains diversity in the solution space and encourages exploration.

***Convergence:*** The algorithm iterates until it converges, i.e., when the improvement in accuracy between generations becomes negligible. At this point, the best weight combination is selected [47].

***Final Weight Selection:*** Once the process is complete, the optimal weight combination $\left(w_{A},w_{B}\right)$ is chosen, resulting in the final prediction ($P_{\mathrm{final}}$ [48]).

#### 3.4.3. Advantages of Cuckoo Search for Model Combination

Cuckoo Search offers several key benefits when it comes to selecting optimal weights for combining deep learning models:

***Efficient Search:*** Thanks to the Lévy flights, CS explores the solution space effectively, covering both small local adjustments and large-scale changes.

***Avoiding Local Optima:*** The algorithm’s ability to replace sub-optimal solutions ensures it does not get stuck in local optima, unlike more traditional methods.

***Simple and Flexible:*** CS is straightforward to implement and can easily be adapted for various optimization problems, including weight selection in deep learning.

Cuckoo Search’s ability to balance exploration and exploitation makes it highly suitable for complex optimization problems like medical image analysis [48].

## 4. Results and Discussions

In the following section, the results of the experiments that were carried out on the model proposed in this study are shown. In addition, we evaluate the outcomes that were achieved by employing a variety of classification strategies within the framework of deep learning models. Python “https://www.python.org/” was used to implement the proposed model, and it was executed on a graphics card with 16 gigabytes of random access memory (RAM) and an RTX 2060 graphics card.

To evaluate the suggested model, we performed multiple experiments using different deep learning models. The design of our model includes three distinct scenarios:Scenario 1: Alzheimer’s disease detection based on deep learning models;Scenario 2: Alzheimer’s disease detection classification based on hybrid deep learning models;Scenario 3: Alzheimer’s disease detection classification based on the weight selection method.

### 4.1. Evaluation Metrics

We utilized standard metrics, including accuracy, precision, recall, and F-measure, to evaluate our categorization method. To evaluate our model’s performance, we used precision, recall, F1-score, ROC-AUC, and Scott’s Pi. Precision focuses on the accuracy of positive predictions, while recall measures the true positive rate. The F1-score combines both, giving a balanced view of how well the model handles false positives and false negatives. Although these metrics are designed for binary classification, they can be adapted to multi-class and ordinal tasks by calculating a micro-average, which treats each class separately.
(7)Accaracy=TP+TN/(TP+TN+FP+FN)
(8)Precision=TP/(TP+FP)
(9)Recall=TP/(TP+FN)
(10)F1−score=2/(1/P+1/R)
*TP* denotes true positive, *FP* denotes false positive, *P* signifies the precision rate, *R* indicates the recall rate, *TPR* represents the true positive rate, and *FPR* signifies the false positive rate.

We also calculated the ROC-AUC score, which gives an overall picture of the model’s performance by considering the balance between true and false positives. Additionally, we used Scott’s Pi, as it provides a more accurate measure for ordinal classification. Scott’s Pi is typically used to assess inter-rater reliability, but it works well for ordinal tasks since it takes the ordered nature of the data into account, making it ideal for our application, according to the research in [49].

### 4.2. Performance Evaluation and Comparative Analysis of Deep Learning Models for Alzheimer’s Disease Detection

The performance of six deep learning models—EfficientNetB0, ResNet50V2, ConvNextBase, InceptionResNetV2, MobileNet1, and EfficientNetV2B3—was evaluated based on their accuracy, validation accuracy, loss, and validation loss. These metrics provide valuable insight into how well each model performed in training and its ability to generalize to unseen data. Table 1 presents the results obtained.

Among the models, ResNet50V2 and EfficientNetV2B3 emerged as top performers, exhibiting the highest accuracy on both training and validation datasets. This indicates that these models not only learned the training data effectively but also maintained strong generalization when applied to new, unseen cases. Their architecture allows them to capture detailed and complex patterns, which is crucial for tasks like Alzheimer’s disease classification, where subtle changes in medical images are key.

On the other hand, MobileNet1, known for being lightweight and efficient, showed lower accuracy compared to the deeper models. While its design favors efficiency, it may not be as effective in handling complex tasks like disease classification. However, MobileNet1 remains a viable option for situations where computational resources are limited or when real-time analysis is required, though at the cost of reduced classification accuracy.

The loss values further help us understand the models’ optimization performance. EfficientNetV2B3 and ResNet50V2, again, proved to be the most effective, showing lower loss and validation loss values, which suggests that these models were able to minimize errors effectively during training and perform consistently across the validation set.

Models like InceptionResNetV2 and ConvNextBase, although showing competitive training loss, had higher validation loss values. This points to potential overfitting, where the model performs well on the training data but struggles to maintain the same performance when evaluated on new data. This discrepancy between training and validation loss is an important consideration, as it affects the model’s reliability in real-world scenarios.

While models such as EfficientNetB0 and MobileNet1 performed reasonably well, their lower validation accuracy and higher loss values suggest that they may have limitations in generalizing to more complex cases. Their simpler architecture is less suited for capturing the intricate features required for precise classifications, particularly in medical imaging. Nonetheless, their computational efficiency makes them attractive options for applications where speed and resource constraints are important, even if some performance is sacrificed.

In the context of Alzheimer’s disease detection, the ability to generalize across different stages of the disease is crucial. Models like EfficientNetV2B3 and ResNet50V2 showed the best balance between high accuracy and low error rates, making them strong candidates for this task. Their superior ability to handle complex image patterns ensures more reliable classifications. On the other hand, models such as MobileNet1 and EfficientNetB0, while efficient, may be more suitable in scenarios where real-time processing or limited computational power is prioritized, even though they might not provide the same level of accuracy.

#### Evaluation of Deep Learning Models for Alzheimer’s Disease Detection Across Multiple Classes

The deep learning models were assessed using key performance metrics such as precision, recall, F1-score, ROC AUC, and Scott’s Pi. These metrics provide insight into both the overall performance of each model and their effectiveness in distinguishing between different stages of dementia. Our results, as detailed in Table 2, demonstrate significant variation in performance metrics.

Overall, EfficientNetB0 and EfficientNetV2B3 emerged as top performers, achieving consistently high scores across all metrics. EfficientNetB0 recorded a micro-average precision, recall, and F1-score of 0.9851, along with an impressive ROC AUC of 0.9995, demonstrating its strong ability to differentiate between Alzheimer’s disease stages. Similarly, EfficientNetV2B3 had near-perfect scores, with a micro-average F1-score of 0.9871 and an ROC AUC of 0.9998, confirming its robust generalization across the dataset. These results indicate that both models excel in terms of accuracy and are highly reliable for clinical use, where distinguishing between different stages of dementia is crucial.

ResNet50V2 also delivered a solid performance, with a micro-average F1-score of 0.9673 and an ROC AUC of 0.9986. However, it slightly lagged behind the EfficientNet models, particularly in its precision and recall values for specific stages. For example, its precision for the non-demented class was 0.9816, slightly lower than 0.9664 forEfficientNetB0, indicating some room for improvement in detecting non-demented cases.

InceptionResNetV2 performed well, with a micro-average F1-score of 0.9524 and an ROC AUC of 0.997. It showed high precision and recall, particularly for the more severe stages of dementia, achieving perfect scores for the mild demented class. However, its slightly lower performance on early-stage detection, like the very mild demented class, where it had an F1-score of 0.9267, suggests that further refinement could enhance its ability to detect the subtler changes associated with early Alzheimer’s disease.

In contrast, ConvNextBase and MobileNet displayed more variability in their results. ConvNextBase performed reasonably well in the moderate demented class, with an F1-score of 0.9123, but it struggled with the mild demented class, achieving a significantly lower F1-score of 0.5899. MobileNet, while more efficient, also struggled with early stages, posting a micro-average F1-score of 0.8845, indicating it may not be as effective at identifying milder cases of Alzheimer’s disease, where distinguishing features are less apparent.

A common challenge for ConvNextBase and MobileNet was their handling of class imbalance. These models tended to perform better in the majority classes, whereas their accuracy dropped for the minority classes, such as the moderate demented category. Addressing this imbalance more effectively, perhaps by using better sampling techniques or model adjustments, would likely enhance their overall performance, particularly in detecting earlier stages of Alzheimer’s disease.

Despite these differences, the ROC AUC values for most models remained high, with even ConvNextBase achieving an ROC AUC of 0.9775, indicating its ability to distinguish between Alzheimer’s disease stages, even though precision and recall may vary between categories.

In conclusion, EfficientNetB0 and EfficientNetV2B3 stood out as the most reliable models for detecting Alzheimer’s disease across different stages thanks to their high accuracy, generalization ability, and near-perfect ROC AUC scores. ResNet50V2 and InceptionResNetV2 also performed well but may benefit from further optimization in early-stage detection. ConvNextBase and MobileNet, while efficient, showed limitations in handling class imbalance and detecting early-stage Alzheimer’s disease, suggesting the need for further refinement to improve their accuracy in these categories.

### 4.3. Alzheimer Detection Based on an Hybrid Deep Learning Model

This evaluation looked at how well different deep learning models—EfficientNetB0, InceptionResNetV2, ResNet50V2, and ConvNeXtBase—performed together by measuring their agreement using Scott’s Pi. This metric helps us understand how well each pair of models aligns in their predictions for Alzheimer’s disease classification, indicating how complementary they are when combined. This evaluation presented in Table 3.

The combination of EfficientNetB0 and InceptionResNetV2 showed the highest agreement, with a Scott’s Pi of 0.98860. This strong result indicates that these two models are highly in sync and likely identify similar patterns in the data. Their consistency makes them a great pairing for improving classification accuracy in an ensemble model.

Similarly, InceptionResNetV2 and ResNet50V2 also demonstrated very high agreement, with a Scott’s Pi of 0.98718. This suggests that these models work well together, providing reliable predictions by capturing complementary features of the dataset. Their combination could be particularly effective in ensuring the robust and accurate detection of Alzheimer’s disease stages. The combination of EfficientNetB0 and ResNet50V2 achieved a Scott’s Pi of 0.98575, which is still strong but slightly lower than the top pairings. This suggests that while these models generally align, there may be minor differences in how they approach the classification task. Nonetheless, they remain a solid combination with a high level of consistency.

The pairing of ResNet50V2 and ConvNeXtBase resulted in a Scott’s Pi of 0.96581. Although this score is lower than others, it still indicates a good level of agreement. This combination might face challenges in consistently identifying certain patterns, but it generally performs well, making it a viable pairing for most classification tasks.

The combination of EfficientNetB0 and ConvNeXtBase scored 0.97507 for Scott’s Pi. While this is a decent level of agreement, it suggests that these two models may have some differences in how they handle certain features, which could affect their overall performance when combined.

The lowest agreement was observed between InceptionResNetV2 and ConvNeXtBase, with a Scott’s Pi of 0.94730. Although these models still perform fairly well together, the lower score indicates more variation in their predictions, likely due to differences in their architectures or how they process the data. This combination may not be as reliable as others for precise classification tasks.

Overall, the combinations of EfficientNetB0 with InceptionResNetV2 and InceptionResNetV2 with ResNet50V2 stood out for their high levels of agreement, making them strong candidates for ensemble models that need reliable and consistent predictions. ConvNeXtBase, while showing reasonable performance, displayed slightly lower alignment with other models, suggesting that it may not be the best choice for pairings when maximum consistency is required. These findings highlight the importance of selecting models that complement each other well, especially when aiming for improved classification accuracy in Alzheimer’s disease detection.

#### Enhanced Alzheimer Detection Through Optimized Weight Selection in Hybrid Deep Learning Models

To further enhance the performance of our model combination, the next step is to refine how we merge the predictions from our best-performing models, EfficientNetV2B3 and InceptionResNetV2. Rather than combining their outputs equally, we will implement a weighted selection method, where each model is assigned a weight based on its accuracy and reliability. This approach allows us to give the model that performs better in specific scenarios more influence, leading to more precise predictions. By optimizing these weights during training, we can create a more balanced and efficient model combination, improving the overall accuracy and making the system more effective at detecting different stages of Alzheimer’s disease. This strategy will ensure that we are using each model’s strengths to their fullest potential.

In this evaluation, we used an optimization algorithm to find the best weight distribution for combining two of our top-performing models, EfficientNetV2B3 and InceptionResNetV2. The aim was to determine how different weight configurations would impact their combined performance, using Scott’s Pi as a measure of agreement between the models. Table 4 presents the results obtained.

The results show that the choice of weights plays a significant role in how well the models work together. When both models were given equal weight (0.5; 0.5), the Scott’s Pi score was 0.9893, indicating strong alignment between the two models. However, we found that slightly favoring EfficientNetV2B3 (with a weight of 0.5; 0.45) actually improved the result, giving us the highest Scott’s Pi score of 0.9907. This suggests that while both models contribute well to the classification, giving EfficientNetV2B3 a bit more influence yields better overall performance.

Other weight combinations, like 0.7; 0.3 and 0.6; 0.45, also resulted in high Scott’s Pi values (0.9864 and 0.9893, respectively), showing that these models are generally quite compatible. However, when the weights were skewed too heavily towards one model, such as 0.2; 0.8 in favor of InceptionResNetV2, Scott’s Pi value dropped significantly to 0.9465. This indicates that putting too much emphasis on one model diminishes the ensemble’s overall effectiveness.

What these results highlight is that the best performance comes from a balanced or near-balanced combination of the two models. EfficientNetV2B3 seems to have a slight edge in handling the data, which is why giving it a bit more weight improves the outcome. The optimization algorithm helped identify this sweet spot and keeps an archive of the best-performing weights, ensuring we can maintain this optimal balance in future runs.

The study demonstrates that combining EfficientNetV2B3 and InceptionResNetV2 works best when their contributions are balanced, with a slight preference for EfficientNetV2B3. The optimal weight distribution of 0.5; 0.45 achieved the highest Scott’s Pi score of 0.9907, confirming that fine-tuning the weights can significantly enhance the models’ agreement and overall performance. This method of optimizing and archiving weights provides a strong foundation for future improvements in model accuracy and reliability.

## 5. Conclusions

This study introduces a novel hybrid deep learning framework designed to enhance the early detection of Alzheimer’s disease through the integration of EfficientNetV2B3 and Inception-ResNetV2 models. By using the Cuckoo Search optimization algorithm, we dynamically adjusted the weighting of these models, achieving a balance (0.5; 0.45) that yielded a Scott’s Pi agreement score of 0.9907, representing near-perfect consistency. This optimized ensemble approach outperformed individual models in terms of accuracy and reliability, particularly in identifying subtle structural changes in the early stages of Alzheimer’s disease. The optimization process not only identified the best-performing configurations but also allowed the system to archive and adapt these configurations for continuous improvement. This adaptability positions the framework as a scalable and practical solution for real-world medical applications. Furthermore, the model’s interpretability techniques provide clinicians with insights into the decision-making process, enhancing its usability and trustworthiness in clinical settings. In conclusion, this optimized hybrid framework represents a significant advancement in AI-assisted medical diagnostics, offering a precise and reliable tool for Alzheimer’s detection. Its ability to detect early-stage Alzheimer’s disease has the potential to improve patient outcomes through timely intervention, paving the way for future developments in neuroimaging technologies and AI-driven healthcare solutions. By addressing a critical need in Alzheimer’s disease diagnosis, this research contributes to a more effective and equitable approach to managing this challenging disease.

## Figures and Tables

**Figure 1 diagnostics-14-02779-f001:**
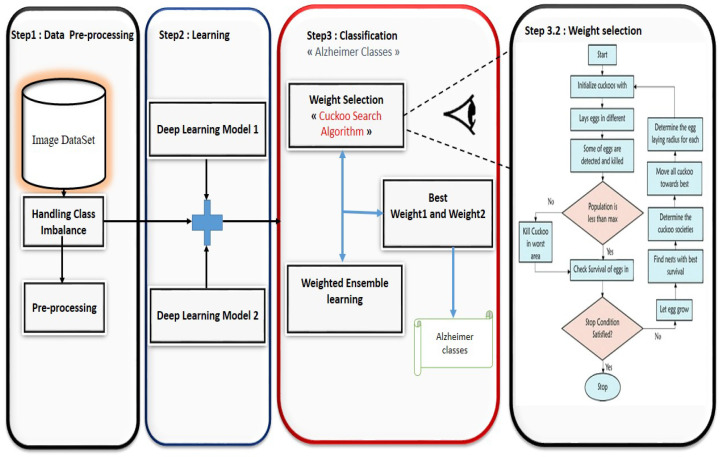
Proposed model for Alzheimer’s disease detection.

**Figure 2 diagnostics-14-02779-f002:**
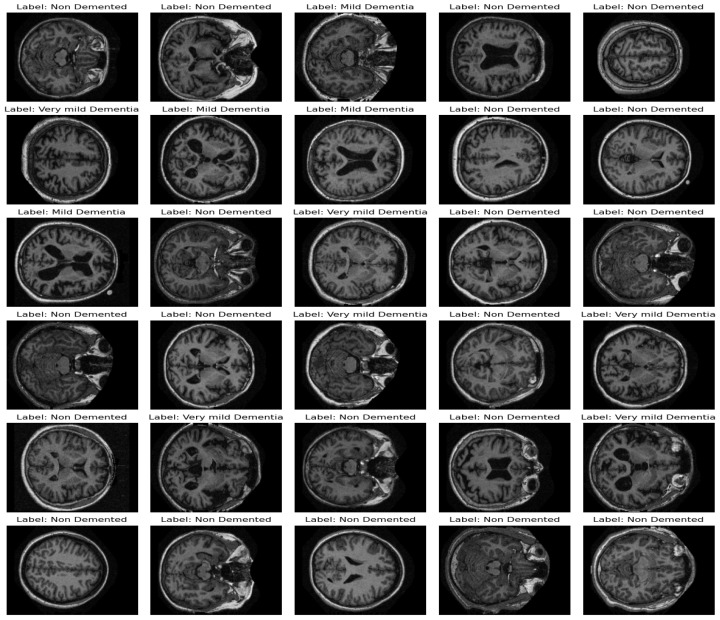
Examples of images taken from the dataset.

**Figure 3 diagnostics-14-02779-f003:**
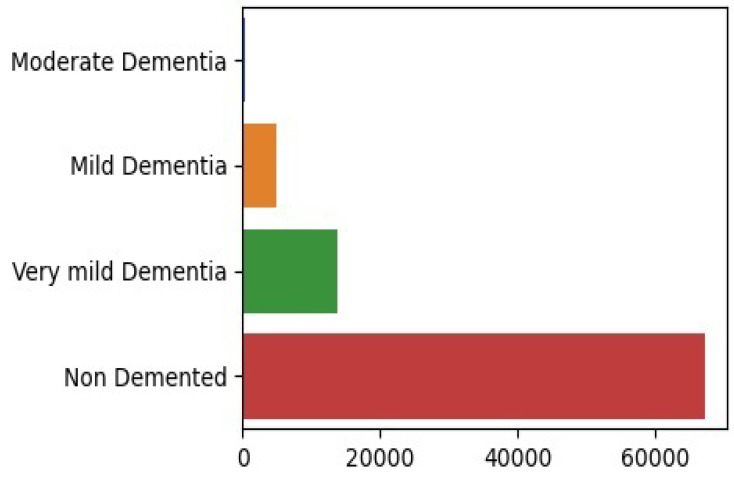
Examples of class imbalance in the dataset.

**Table 1 diagnostics-14-02779-t001:** Performance accurracy of different deep learning models.

Deep Learning Model	Accuracy	Loss
EfficietNetB0	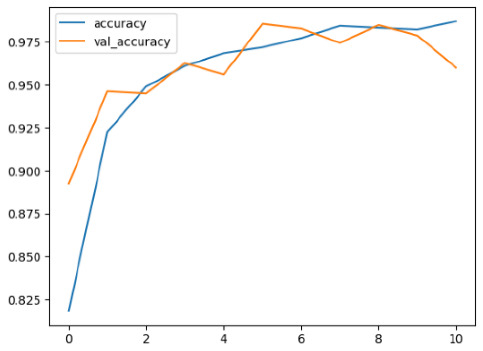	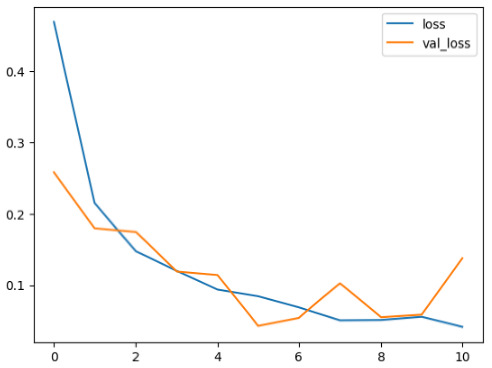
ResNet50V2	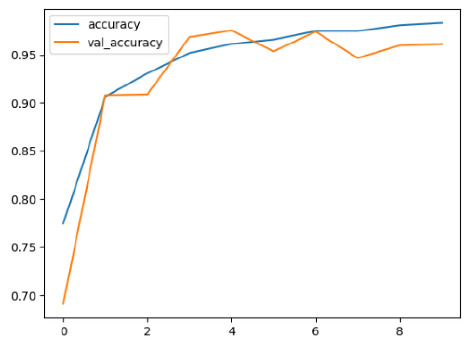	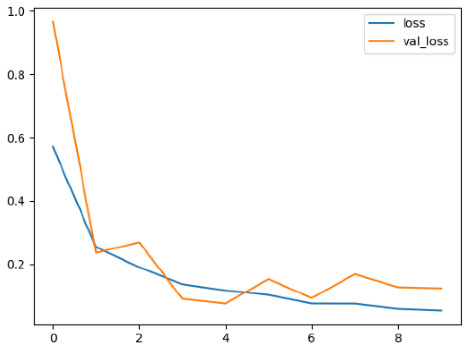
ConvNextBase	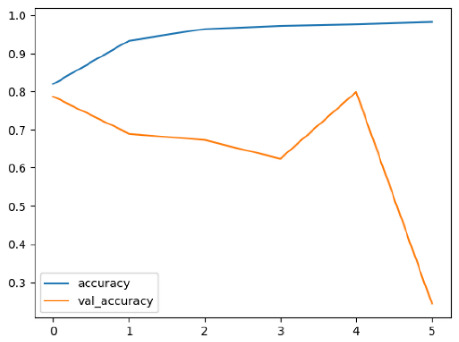	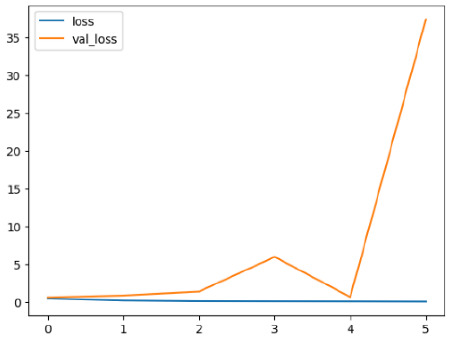
InceptionResNetV2	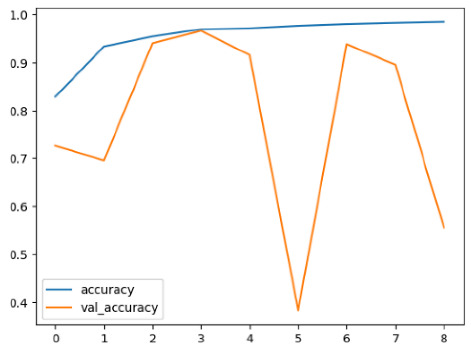	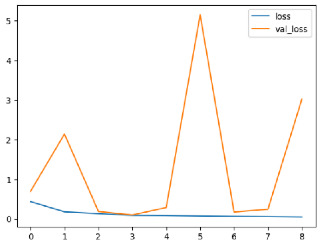
MobilNet	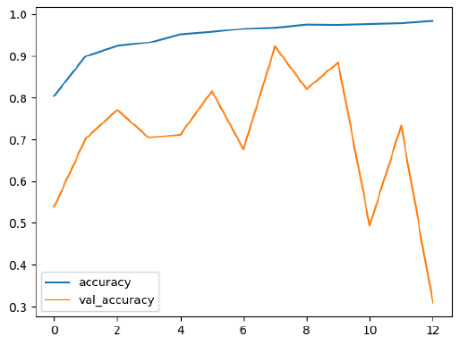	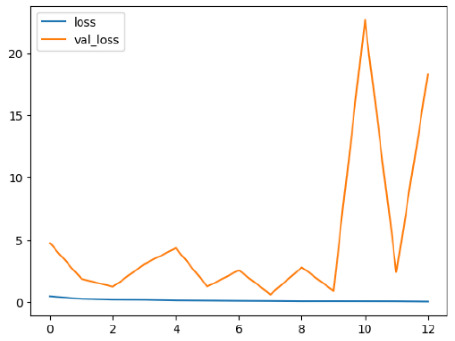
EfficietNetV2B3	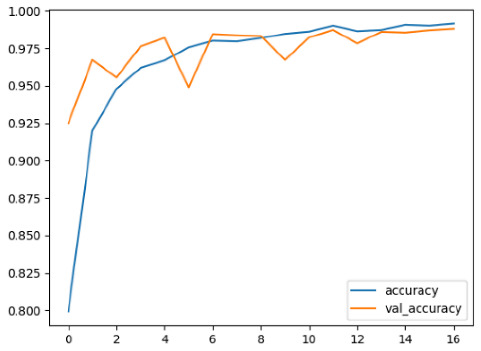	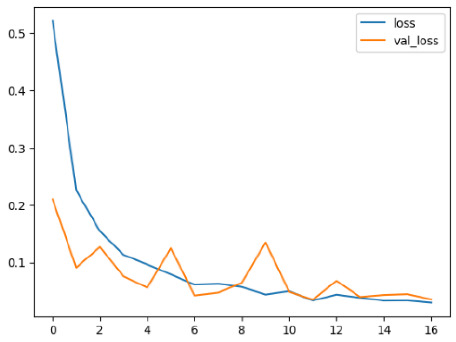

**Table 2 diagnostics-14-02779-t002:** Performance metrics for different deep learning models.

	Class Names	Precision	Recall	F1-Score	ROC AUC	Scott’s Pi
**EfficientNetB0**	* **non_demented** *	0.9664	0.9891	0.9776	0.9988	–
	* **very_mild_demented** *	0.9920	0.9672	0.9794	0.9991	–
	* **moderate_demented** *	0.9953	0.9969	0.9961	1.0000	–
	* **mild_demented** *	1.0000	1.0000	1.0000	1.0000	–
	* **Micro-Average** *	0.9851	0.9851	0.9851	0.9995	0.97863
**ResNet50V2**	* **non_demented** *	0.9816	0.9156	0.9475	0.9963	–
	* **very_mild_demented** *	0.9291	0.9828	0.9552	0.9966	–
	* **moderate_demented** *	0.9892	0.9984	0.9938	0.9999	–
	* **mild_demented** *	1.0000	1.0000	1.0000	1.0000	–
	* **Micro-Average** *	0.9673	0.9673	0.9673	0.9986	0.95300
**ConvNextBase**	* **non_demented** *	0.7977	0.8688	0.8317	0.9624	–
	* **very_mild_demented** *	0.9381	0.7578	0.8384	0.9822	–
	* **moderate_demented** *	0.8388	1.0000	0.9123	0.9995	–
	* **mild_demented** *	1.0000	0.4184	0.5899	1.0000	–
	* **Micro-Average** *	0.8533	0.8533	0.8533	0.9775	0.78689
**InceptionResNetV2**	* **non_demented** *	0.8996	0.9656	0.9314	0.9936	–
	* **very_mild_demented** *	0.9677	0.8891	0.9267	0.9944	–
	* **moderate_demented** *	0.9876	0.9953	0.9914	0.9999	–
	* **mild_demented** *	1.0000	1.0000	1.0000	1.0000	–
	* **Micro-Average** *	0.9524	0.9524	0.9524	0.997	0.931639
**MobilNet**	* **non_demented** *	0.9279	0.8250	0.8734	0.9714	–
	* **very_mild_demented** *	0.9277	0.8219	0.8716	0.9829	–
	* **moderate_demented** *	0.8384	0.9891	0.9075	0.9928	–
	* **mild_demented** *	0.7717	1.0000	0.8711	0.9974	–
	* **Micro-Average** *	0.8845	0.8845	0.8845	0.9814	0.83499
**EfficietNetV2B3**	* **non_demented** *	0.9889	0.9750	0.9819	0.9993	–
	* **very_mild_demented** *	0.9798	0.9844	0.9821	0.9994	–
	* **moderate_demented** *	0.9907	1.0000	0.9953	1.0000	–
	* **mild_demented** *	1.0000	1.0000	1.0000	1.0000	–
	* **Micro-Average** *	0.9871	0.9871	0.9871	0.9998	0.98148

**Table 3 diagnostics-14-02779-t003:** Evaluation of Average ensemble learning model for Alzheimer’s disease detection.

Deep Learning Model	InceptionResNetV2	ResNet50V2	ConvNeXtBase
**EfficientNetB0**	**0.98860 **	0.98575	0.97507
**InceptionResNetV2 **		0.98718	0.94730
**ResNet50V2 **			0.96581

**Table 4 diagnostics-14-02779-t004:** Performance metrics of optimized weighted average models for Alzheimer’s disease detection using optimized selected alpha (α) and Beta (β) values.

Ensemble Learning Model	Weight	Scott’s Pi
**EfficientNetV2B3** **+** **InceptionResNetV2**	(0.8; 0.2)	0.9829
	(0.9; 0.1)	0.9829
	(0.7; 0.3)	0.9864
	(0.5; 0.5)	0.9893
	**(0.5; 0.4) **	**0.9900 **
	(0.4; 0.6)	0.9672
	(0.2; 0.8)	0.9465
	**(0.5; 0.45) **	**0.9907 **
	(0.6; 0.45)	0.9893
	(0.5; 0.3)	0.9888

## Data Availability

The data used in this study are openly accessible at the following link: https://www.oasis-brains.org/ (accessed on 15 August 2024).

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
