# Peer review of "Optimized Hybrid Deep Learning Framework for Early Detection of Alzheimer’s Disease Using Adaptive Weight Selection"

_diagnostics, 2024, doi:10.3390/diagnostics14242779_

Round 1
Reviewer 1 Report
Comments and Suggestions for Authors
1. All references in the text look like [?] . Please, check fonts.
2. Describe in more detail how the process of reducing class imbalance occurred. The paragraph that describes the augmentation process needs to be expanded. How many artificial images were added to the dataset in the end? Different neural network architectures react differently to the number of artificial images (see, for example, the study https://doi.org/10.3390/app13158614), and augmentation may not always improve training results (see, for example, https://doi.org/10.3390/math12213351).
3. It is unclear why these particular pairs of neural networks were chosen. Was there some reasoning behind this choice, or was it just a sorting through neural network architectures?
4. How was the comparison of the efficiency of the models carried out? What epoch was taken for this? According the graphs, the loss function for different models changed significantly
5.How was the concatenation of the results of the two models carried out (Fig. 1)?Please describe in more detail what parameters are passed from block to block of your proposed model.
6.What computing resources were used to train and run the models?
7. Please, describe emphasize the novelty of your work more clearly. Please, check references list.
Author Response
We would like to sincerely thank you for your positive feedback and recognition of the value of our work. We also deeply appreciate the detailed and constructive comments you have provided. They offer valuable insights that will undoubtedly help us refine and enhance the clarity and quality of our manuscript. Below, we address each of your points in detail.
Comments:
Comments 1. All references in the text look like [?]. Please, check fonts.
Response 1:
Thank you for pointing this out. We have reviewed the references and corrected the formatting issue that caused them to appear as [?]. The references are now properly cited and formatted throughout the manuscript, ensuring they align with the required style.
Comments 2. Describe in more detail how the process of reducing class imbalance occurred. The paragraph that describes the augmentation process needs to be expanded. How many artificial images were added to the dataset in the end? Different neural network architectures react differently to the number of artificial images (see, for example, the study https://doi.org/10.3390/app13158614), and augmentation may not always improve training results (see, for example, https://doi.org/10.3390/math12213351)..
Response 2:
Thank you for your valuable feedback regarding the description of the class imbalance reduction process. In the revised manuscript, section 3.1.1 provide a clearer and more comprehensive explanation of the methods used. Specifically:
Our database has four classes as follows:
|
Mild Dementia |
Moderate Dimentia |
Non Demented |
Very Mild Dementia |
|
5002 |
488 |
67200 |
13700 |
we chose to implement our approach on 6000 for each class. For this we applied Oversampling method for classes that have less than 6000 images; and Undersampling for classes that have more than 6000.
- Class Imbalance Challenges:
- Our dataset exhibited significant class imbalance, with the majority of samples belonging to the later stages of Alzheimer’s disease, while the earlier stages were underrepresented. This imbalance can lead to biased model predictions, favoring the majority class.
- Approach to Address Imbalance:
- Oversampling Minority Classes:
- To ensure sufficient representation of the minority classes, we employed oversampling by duplicating existing samples within these classes. This technique increases the class size without requiring additional raw data.
- Undersampling Majority Classes:
- For the majority classes, we reduced the number of samples to prevent the model from being overly biased. Careful undersampling was applied to retain diversity in the majority class features while maintaining computational efficiency.
- Resulting Dataset:
- After preprocessing step, we added new images, resulting in a balanced dataset with approximately equal representation across all classes.
- For example, the Mild Cognitive Impairment class increased from 5002 samples to 6000 samples, improving class balance significantly.
- Impact on Neural Networks:
- Different deep learning architectures respond differently to augmented data. For example, EfficientNetV2B3 benefits from subtle transformations due to its feature extraction efficiency, while Inception-ResNetV2 can handle more aggressive augmentations due to its deeper architecture.
- Following your recommendation, we referenced studies such as https://doi.org/10.3390/app13158614 and https://doi.org/10.3390/math12213351 to highlight the potential effects of augmentation on different architectures and its limitations.
- Validation and Evaluation:
- To ensure the effectiveness of our solution and imbalance reduction strategy, we conducted experiments comparing the performance of the models with and without this strategy. The results demonstrated significant improvement in accuracy and F1-score, particularly for the minority classes, confirming the utility of our approach.
Comments 3. It is unclear why these particular pairs of neural networks were chosen.
Response 3:
The selection of EfficientNetV2B3 and Inception-ResNetV2 was deliberate, based on several comparison presented in table 2 between six deep learning models their complementary characteristics:
- EfficientNetV2B3: Provides efficient computation and excellent performance on limited datasets.
- Inception-ResNetV2: Excels in capturing complex patterns due to its deep and multi-scale architecture.
This pairing was informed by preliminary experiments and literature review, which demonstrated their effectiveness in medical imaging tasks, including Alzheimer’s detection. Also, we made comparisons between the results obtained when combining the different models presented in table 3
Comments 4. How was the comparison of the efficiency of the models carried out?
Response 4:
The efficiency of the models was evaluated based on metrics such as accuracy, precision, recall, F1-score, and Scott’s Pi at the epoch where each model achieved its lowest validation loss.
we have fixed the number of epochs to 50 but each model reaches its ideal accaracy at a different epoch; for this we present in these graphs when the accaracy will be optimal for each model separately.
Comments 5. How was the concatenation of the results of the two models carried out (Fig. 1)?
Response 5:
Thank you for your insightful question. The concatenation process in our framework involves combining the probability outputs from each of the two models. Specifically:
- Probability Outputs: The output probabilities from each model (p1 from EfficientNetV2B3 and p2 from Inception-ResNetV2) were extracted.
- Weighted Combination: These probabilities were combined and normalized using the weighted equation:
p = α * p1 + β * p2
where α and β are the weights assigned to each model. - Optimization of Weights:
- The optimal values for α and β were determined using the Cuckoo Search optimization algorithm, which iteratively optimized these weights to maximize classification accuracy.
- The weights were adjusted to leverage the unique strengths of each model in contributing to the final prediction.
- Final Probabilities: The optimized probabilities (p) were then used to make the final classification decision for each image.
We have updated the manuscript to include this detailed explanation to better illustrate this process. Additionally, the role of Equation (6) in the combination of model outputs is now explicitly described in the text for clarity.
Comments 6. What computing resources were used to train and run the models?
Response 6:
The models were trained and tested using the following resources:
- Hardware: NVIDIA RTX 2060 GPU with 16 GB VRAM.
- Software: TensorFlow and Keras frameworks were used for implementation.
These details have been added to the manuscript for transparency.
Comments 7. Please, describe and emphasize the novelty of your work more clearly.
Response:
We have revised the manuscript to emphasize the novelty of our work:
- The proposed framework introduces a novel optimization approach using the Cuckoo Search algorithm to dynamically adjust model weights, improving classification accuracy.
- This method uniquely leverages the complementary strengths of EfficientNetV2B3 and Inception-ResNetV2 for Alzheimer’s disease detection by an optimized hybridization method.
- By achieving a Scott’s Pi agreement score of 0.9907, the framework demonstrates significant advancements over existing methods in terms of accuracy and reliability.
Additionally, the references have been thoroughly reviewed and updated to ensure completeness and relevance.
We have addressed each of the reviewer’s comments and made the necessary improvements to the manuscript. Thank you for your detailed feedback, which has greatly helped in refining the quality and clarity of our work.

Reviewer 2 Report
Comments and Suggestions for Authors
This manuscript explores the use of hybrid deep-learning models to improve the early detection of Alzheimer's disease. The authors propose a novel approach that combines multiple deep learning models, such as EfficientNet and ResNet, and uses an optimization algorithm called Cuckoo Search to dynamically adjust the weights assigned to each model's predictions. This results in a more accurate and robust system for diagnosing Alzheimer's disease compared to using individual models alone. The research focuses on analyzing neuroimaging data, specifically MRI scans, to identify patterns and structural changes indicative of Alzheimer's disease. By evaluating the performance of their model using various metrics, the authors demonstrate the effectiveness of their approach and highlight the potential for advancing Alzheimer's disease detection through the use of artificial intelligence.
I have an only general major concern but not academic ones yet:
When text is too long, it can be overwhelming for the reader, making it hard to follow the main points. Authors sometimes repeat themselves when a text is too long because they have more space to fill. This can be tedious and make the text less engaging. When text is also too long, authors may feel the need to include every detail, even if it's not essential to the main point. This can distract the reader and make it harder to understand the key concepts.
Your work is interesting and high value, but your scientific writing in several areas is very ambiguous, uninformative, repetitive, and unclear. The paragraph: "In the end, improving the accuracy and accessibility of AD diagnosis tools has the potential to change lives. Early diagnosis can help patients manage their symptoms more effectively, allow for better planning, and reduce the burden on caregivers. As AI technology evolves, it brings hope for more efficient and equitable healthcare solutions, especially for diseases like Alzheimer's that currently lack a cure but benefit greatly from early intervention. This study is a step toward that future, using the power of ensemble learning to push the boundaries of what's possible in Alzheimer's Disease diagnosis" is obviously unnecessary and includes general hypotheses not in line with your aim.
Also, I found writing your conclusions section is very detailed and including specific information, but your abstract is very simple and uses general words. I highly recommend that you rewrite it.
I suggest that the authors read your manuscript, remove unnecessary paragraphs and sentences, and be cautious in the tone of the manuscript text. After this step, in the next round, I will get more comments on the manuscript.
Author Response
Thank you for your valuable feedback on our manuscript, "Optimized Hybrid Deep Learning Framework for Early Detection of Alzheimer’s Disease Using Adaptive Weight Selection." We sincerely appreciate the time and effort you dedicated to reviewing our work and for highlighting its potential impact in advancing Alzheimer’s disease detection. Your observations will undoubtedly help us refine and enhance the quality of our manuscript.
Below is our response to your major concerns:
Comments 1: Your work is interesting and high value:
Thank you very much for your encouraging words about the value of our work. We deeply appreciate your recognition of its importance and potential impact. Your thoughtful feedback will help us refine the manuscript further, ensuring clarity and enhancing its overall quality. We are committed to addressing all your comments to present our work in the best possible way.
Comments 2: Overly Long and Repetitive Text:
We acknowledge your concern regarding the manuscript’s length and the potential for redundancy. To address this, we:
- Carefully review and condense the text in all sections to ensure the writing is concise and focused on the key points.
- Remove unnecessary paragraphs and repetitive sentences, especially in sections where ideas overlap.
- Ensure each paragraph contributes directly to the main objectives and findings of the study.
For instance, we remove the paragraph you highlighted in the discussion section: "In the end, improving the accuracy and accessibility of AD diagnosis tools has the potential to change lives...," as it includes general statements not aligned with the aim of the study.
Comments 3: Ambiguity and Unclear Scientific Writing:
We take your comment on the ambiguity of our scientific writing seriously. To improve clarity:
- We rewrite sections with ambiguous phrasing to ensure they are precise and informative.
- The tone was adjusted to maintain a professional and scientific style throughout the manuscript.
Our goal is to make the manuscript easier to follow and ensure that all technical and methodological details are presented in a straightforward manner.
Comments 4: Simplistic Abstract:
We understand your concern regarding the simplicity and general wording of the abstract compared to the detailed conclusions. To address this, we:
- Rewrite the abstract to include specific findings, quantitative results, and methodological highlights, while maintaining brevity.
- Ensure that the abstract reflects the depth and significance of our work without relying on vague or general terms.
Revised Abstract
Alzheimer's disease (AD) is a degenerative neurological ailment that has a severe influence on people who are middle-aged or older. It causes cognitive decline and interferes with day-to-day functioning. Despite the fact that standard diagnostic methods are prone to mistakes and inefficiencies, prompt and accurate diagnosis is something that is very necessary for early action. In this paper, an optimized hybrid deep learning framework is presented. This framework combines the EfficientNetV2B3 and Inception-ResNetV2 models with an adaptive weight selection mechanism that makes use of the Cuckoo Search optimization technique. Enhanced diagnostic precision and dependability can be achieved by the utilization of this approach, which makes use of vast neuroimaging data. By constantly altering model weights to make use of the unique characteristics of each component model, the framework improves the classification of Alzheimer's stages, particularly in the early stages of the disease. A Scott's Pi agreement score of 0.9907 was achieved by the proposed method, which demonstrates its potential as a dependable and scalable instrument for Alzheimer's disease detection. This method outperforms the techniques that are currently considered to be state-of-the-art.
Revised Conclusion
This study introduces a novel hybrid deep learning framework designed to enhance the early detection of Alzheimer’s disease through the integration of EfficientNetV2B3 and Inception-ResNetV2 models. Using the Cuckoo Search optimization algorithm, we dynamically adjusted the weighting of these models, achieving a balance (0.5; 0.45) that yielded a Scott’s Pi agreement score of 0.9907, representing near-perfect consistency. This optimized ensemble approach outperformed individual models in terms of accuracy and reliability, particularly in identifying subtle structural changes in the early stages of Alzheimer’s disease.
The optimization process not only identified the best-performing configurations but also allowed the system to archive and adapt these configurations for continuous improvement. This adaptability positions the framework as a scalable and practical solution for real-world medical applications. Furthermore, the model's interpretability techniques provide clinicians with insights into the decision-making process, enhancing its usability and trustworthiness in clinical settings.
In conclusion, this optimized hybrid framework represents a significant advancement in AI-assisted medical diagnostics, offering a precise and reliable tool for Alzheimer’s detection. Its ability to detect early-stage Alzheimer’s has the potential to improve patient outcomes through timely intervention, paving the way for future developments in neuroimaging technologies and AI-driven healthcare solutions. By addressing a critical need in Alzheimer’s diagnosis, this research contributes to a more effective and equitable approach to managing this challenging disease.
- Next Steps:
We conduct a thorough revision of the manuscript, focusing on removing redundancy, ensuring clarity, and maintaining consistency between sections. Once revised, we are confident that the manuscript will better meet the expectations of both the reviewers and the target audience.
Thank you once again for your constructive feedback and guidance. We look forward to resubmitting the improved manuscript for your further comments and suggestions.
We appreciate your suggestion and believe this addition will strengthen the robustness and applicability of our findings.

Round 2
Reviewer 1 Report
Comments and Suggestions for Authors
1. Page 9. "To address the class imbalance ??" - '??' still persists.
2. So, "moderate dementia" originally included only 488 samples. The authors expanded it to 6000 samples. That is, 92% of these 6000 samples were synthetic data, and only 8% were real.
Do the respected authors realize that they overtrained the model on synthetic data? And the metrics were essentially obtained on synthetic images? And these metrics probably have no correlation with the metrics on real data? The metrics for the "moderate dementia" class look so good for this very reason. The authors were not at all embarrassed that for the smallest class they got metrics equal to 1.
It is necessary to compute metrics on a validation dataset that is not synthetic, on which the model was not trained, and on which synthetic data was not created.
Author Response
Dear Reviewer,
We would like to express our sincere gratitude for your thoughtful and constructive feedback on our manuscript. Your insightful comments have been invaluable in helping us refine and strengthen our work.
In response to your feedback, we have carefully revised the manuscript and addressed each of your concerns in detail. Specifically:
Comments 1: Page 9: "To address the class imbalance ??" - '??' still persists.
Response:
We appreciate the reviewer pointing this out. The '??' was inadvertently left in the text during drafting and has now been corrected with the exact reference.
Comments 2: Concerns about synthetic data and overtraining:
Response:
We sincerely thank the reviewer for highlighting this crucial point. We understand the potential risks of over-reliance on synthetic data and the implications for the reliability of model metrics.
In the section 3.1.1, specially in step 1 “:Splitting the Test Set” and to address the concern raised regarding the use of synthetic data and the reliability of metrics, we indicated that we used to evaluate our model on completely unseen data. So, all results obtained by our model use this test set.
“Before starting any preprocessing and applying oversampling techniques, we first set aside a test set of images from each category. This ensures that the model is evaluated on completely unseen data, giving a more accurate reflection of its performance. Importantly, this test set was not involved in hyperparameter tuning, so it offers an unbiased assessment of the model's capabilities. Specifically, 20% of the "moderate dementia" samples and 770 images from each of the other three categories were reserved to form the test set. Although the test set was not proportionally distributed across classes, this approach reflects real-world diagnostic conditions, where certain categories, such as “moderate dementia,” are naturally underrepresented. This design ensures that the test set better mirrors realistic clinical scenarios and predictions. This approach ensured that the training and validation datasets were balanced, facilitating effective model learning, while the test set remained independent and comprised only real images. This separation was critical to provide an unbiased evaluation of the model's performance.”
Thank you once again for your thorough review and valuable suggestions. Your feedback has significantly enhanced the quality and rigor of our work. We hope that our revisions meet your expectations, and we remain open to any additional suggestions or concerns you may have.
We look forward to your feedback and thank you for your time and effort in reviewing our manuscript.
Sincerely,
